# Effect and safety of ethanolamine oleate in sclerotherapy in patients with difficult-to-resect venous malformations: A multicenter, single-arm study

**Mine Ozaki** [1]*, **Tadashi Nomura** [2], **Keigo Osuga** [3], **Masakazu Kurita** [4], **Ayato Hayashi** [5,6], **Shunsuke Yuzuriha** [7], **Noriko Aramaki-Hattori** [8], **Makoto Hikosaka** [9], **Taiki Nozaki** [10], **Michio Ozeki** [11], **Junko Ochi** [12], **Shimpei Akiyama** [13], **Yasumasa Kakei** [14,15], **Keiko Miyakoda** [15], **Naoko Kashiwagi** [15], **Takahiro Yasuda** [16,17], **Yuki Iwashina** [1], **Tsuyoshi Kaneko** [9], **Kiyoko Kamibeppu** [18], **Takafumi Soejima** [19], **Kiyonori Harii** [1]

1 Department of Plastic Surgery, Kyorin University Faculty of Medicine, Tokyo, Japan, 2 Department of Plastic Surgery, Kobe University Graduate School of Medicine, Kobe, Hyogo, Japan, 3 Department of Diagnostic Radiology, Osaka Medical and Pharmaceutical University, Osaka, Japan, 4 Department of Plastic and Reconstructive Surgery, The University of Tokyo Hospital, Tokyo, Japan, 5 Department of Plastic and Reconstructive Surgery, Yokohama City University, Yokohama, Kanagawa, Japan, 6 Department of Plastic and Reconstructive Surgery, Juntendo University School of Medicine, Tokyo, Japan, 7 Department of Plastic and Reconstructive Surgery, Shinshu University School of Medicine, Matsumoto, Nagano, Japan, 8 Department of Plastic and Reconstructive Surgery, Keio University School of Medicine, Tokyo, Japan, 9 Department of Plastic and Reconstructive Surgery, National Center for Child Health and Development, Tokyo, Japan, 10 Department of Radiology, Keio University School of Medicine, Tokyo, Japan, 11 Department of Pediatrics, Graduate School of Medicine, Gifu University, Gifu, Japan, 12 Department of Diagnostic Radiology, Suita Tokushukai Hospital, Osaka, Japan, 13 Department of Radiology, Kyoto Prefectural University of Medicine, Kyoto, Japan, 14 Department of Oral and Maxillofacial Surgery, Kobe University Graduate School of Medicine, Kobe, Hyogo, Japan, 15 Clinical and Translational Research Center, Kobe University Hospital, Kobe, Hyogo, Japan, 16 Department of Medical Devices, Kobe University Graduate School of Medicine, Kobe, Hyogo, Japan, 17 Advanced Medical-Engineering Development Center, Kobe University, Kobe, Hyogo, Japan, 18 Division of Health Sciences and Nursing, Graduate School of Medicine, the University of Tokyo, Tokyo, Japan, 19 Department of Child Health Nursing, Kobe University Graduate School of Health Sciences, Kobe, Hyogo, Japan

* zakimin@nifty.com

**Data Availability Statement:** All relevant data are within the paper and its Supporting Information files.

## Abstract

### Objective

To evaluate the effect and safety of sclerotherapy in patients with difficult-to-resect venous malformations treated with ethanolamine oleate.

### Design and setting

This investigator-initiated clinical trial employed a multicenter, single-arm design and was conducted in Japan.

### Patients

Overall, 44 patients with difficult-to-resect venous malformations were categorized into two cohorts: 22 patients with cystic-type malformations and 22 patients with diffuse-type malformations, including children (<15 years old).

**Funding:** This research was supported by the Japan Agency for Medical Research and Development (AMED), Grant Number JP20lk0201115. The funders had no role in study design, data collection and analysis, decision to publish, or preparation of the manuscript.

**Competing interests:** The authors have read the journal's policy and have the following competing interests: Fuji Chemical Industries Co., Ltd. provided ethanolamine oleate for treatment during the study period. There are no patents, products in development or marketed products associated with this research to declare. This does not alter our adherence to PLOS ONE policies on sharing data and materials.

## Interventions

Adult patients received injections of 5% ethanolamine oleate solution, double diluted with contrast or normal saline, with a maximum dose of 0.4 mL/kg. The same method of administration was used for children (<15 years old). The maximum volume of the prepared solution in one treatment was 30 mL.

## Evaluation methods

Treatment effect was assessed by evaluating the difference in lesion volume using magnetic resonance imaging as a primary endpoint and differences in pain using a visual analog scale as a key secondary endpoint.

## Results

Among the 45 patients who consented, one was excluded owing to potential intracranial involvement of venous malformations during screening. Regarding the primary outcome, 26 of 44 patients (59.1%, 95% confidence interval: 44.41–72.31%) achieved $\geq$ 20% reduction in malformation volume, with 16 patients having cystic lesions (72.7%, 51.85–86.85%) and 10 patients having diffuse lesions (45.5%, 26.92–65.34%). Both cohorts showed significant improvement in self-reported pain scores associated with lesions 3 months post-sclerotherapy. No death or serious adverse events occurred. Hemoglobinuria was observed in 23 patients (52%), a known drug-related adverse event. Prompt initiation of haptoglobin therapy led to full recovery within a month for these patients.

## Conclusions

Ethanolamine oleate shows potential as a therapeutic sclerosing agent for patients with difficult-to-resect venous malformations.

## Introduction

Venous malformations (VMs) are vascular developmental abnormalities that arise during fetal life and are associated with pain, movement disorders, and other symptoms [1–3]. A study conducted by the Ministry of Health, Labor and Welfare in Japan (Project on Intractable Hemangioma, Vascular Malformation, Lymphangioma, Lymphangiomatosis and Related Diseases in Fiscal 2014) estimated that 20,000 individuals are affected by VMs, with half of them harboring difficult-to-resect lesions due to extensive involvement or muscle infiltration [4, 5]. For such cases, sclerotherapy is the treatment of choice [2, 6–9].

Sclerotherapy is a globally recognized treatment for VMs, described as among the most effective options available. Ethanolamine oleate, a sclerosing agent, has been extensively documented in this regard [10]. American guidelines for percutaneous sclerotherapy for head and neck venous and lymphatic malformations [11] and European guidelines for sclerotherapy in chronic venous disorders both acknowledge the effectiveness of sclerotherapy in chronic venous disorders, including VMs [12]. However, to the best of our knowledge, sclerotherapy for VMs remains uncovered by health insurance policies, with no pharmaceutical approval for any sclerosing agent across countries to date.

In Japan, ethanolamine oleate, absolute ethanol, and polidocanol have been used as off-label sclerosing agents for VM treatment [13–17]. While absolute ethanol and polidocanol are associated with serious, life-threatening adverse events [11, 18, 19], ethanolamine oleate is considered a safer option with minimal adverse events [20]. Horbach et al.'s article [20], which analyzed five studies encompassing 188 patients treated with ethanolamine oleate for venous or lymphatic malformations, reported skin ulceration and skin necrosis (six cases, 3%) as adverse events. However, systemic adverse events, facial nerve disorders, or other adverse events were absent [20]. In another study that evaluated the cytotoxicity of each sclerosing agent in muscle tissue, absolute ethanol was the most cytotoxic, with ethanolamine oleate and polidocanol exhibiting comparably lower cytotoxicity [14]. Conversely, studies have shown promising results for ethanolamine oleate, which reduced lesions and improved symptoms in 88–100% of cases evaluated in a systematic review [20]. Furthermore, a pediatric-centered study examined its application in children (mean age: 15.1 years, range: 3 months to 21 years) among 83 patients (comprising 85 procedures), resulting in complete symptom resolution for 79 lesions and notable enhancements for six lesions [21]. Although ethanolamine oleate is currently approved as a sclerosing agent for esophageal and gastric varices in Japan [22, 23], its use has not been indicated for pediatric patients. This is particularly relevant considering the congenital nature of VMs, often requiring treatment in children. Nevertheless, existing evidence suggests its effect in pediatric patients.

Given its effect and safety profile, ethanolamine oleate is regarded as the most appropriate candidate for initial regulatory approval as a sclerosing agent for VMs. A limited number of well-designed clinical trials have investigated treatment options for difficult-to-resect VMs; we initiated a phase III, multicenter, single-arm clinical trial to aid the Japanese government in approving health insurance coverage.

Based on their morphological characteristics, VMs can be broadly categorized into cystic and diffuse lesions. Cystic lesions are more likely to shrink with sclerotherapy, whereas diffuse lesions are less likely to respond [24]. Consequently, the study design employed volume reduction and pain improvement as the evaluation criteria for cystic and diffuse lesions, respectively. This clinical trial aimed to characterize the clinical impact of ethanolamine oleate injection in patients with difficult-to-resect VMs.

## Materials and methods

The protocol for this trial and TREND Statement Checklist are available as supporting information; see S1–S3 Protocols and S1 File.

### Study design and procedure

This non-randomized, prospective, single arm, multicenter clinical trial was conducted at eight hospitals from January 1, 2021 to the end of April 2023 (the last patient's visit). The recruitment period for Kyorin University Hospital, Kobe University Hospital, Osaka Medical and Pharmaceutical University Hospital, The University of Tokyo Hospital, Juntendo University Urayasu Hospital, and National Center for Child Health and Development Hospital was from 1/1/2021 to 31/3/2023; for Shinshu University Hospital, the recruitment period was from 1/2/2021 to 31/3/2023; for Keio University Hospital, the recruitment period was from 19/3/2021 to 31/3/2023. A summary of this study is presented in Fig 1. The observation period continued for 3 months following the intervention. Participants were instructed to visit at designated times during the observation period and provided with reimbursement for travel expenses to reduce the burden on patients. Written informed consent was obtained from all patients or their parents/guardians if the patient was under 20 years old. This clinical trial was

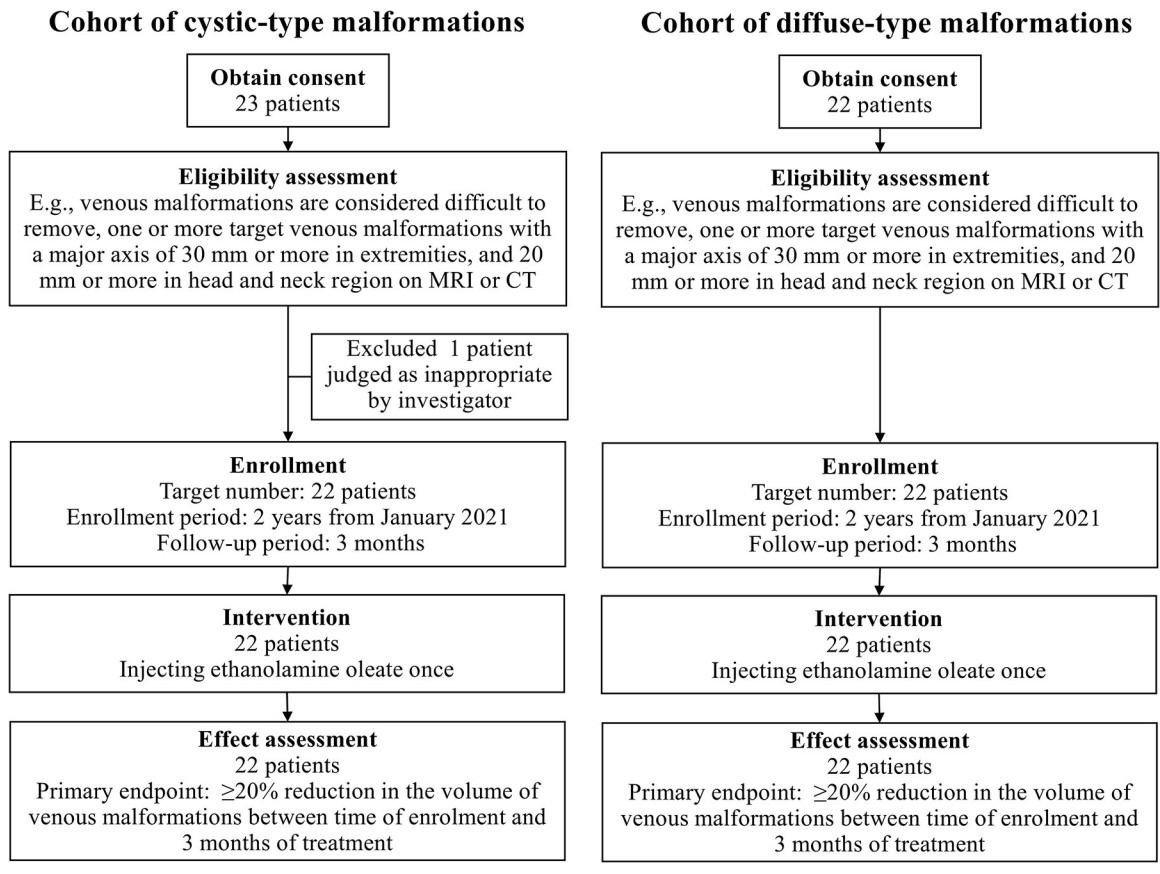

**Fig 1. Flow diagram describing participation in this study.** MRI, magnetic resonance imaging; CT, computed tomography.

approved by the Institutional Review Board of each facility (Kyorin University Hospital, October 14, 2020, Reference number 2017, Kobe University Hospital, July 15, 2020, Reference number 200016, Osaka Medical and Pharmaceutical University Hospital, November 17, 2020, Reference number 20-1-21-0462DR, Shinshu University Hospital, January 26, 2021, Reference number 1852, The University of Tokyo Hospital, November 16, 2020, Reference number 202021-11DX, Keio University Hospital, February 24, 2021, Reference number D20-05, Juntendo University Urayasu Hospital, September 15, 2020, Reference number 20-04-227, National Center for Child Health and Development Hospital, November 19, 2020, Reference number C32002). This study was conducted using good clinical practice and adhered to the principles of the Declaration of Helsinki. It was registered with the Japan Registry of Clinical Trials (Identifier: jRCT2051200046) under the recommendations of the International Committee of Medical Journal Editors (Registered on August 25, 2020). (https://jrct.niph.go.jp/latest-detail/jRCT2051200046). The study protocol adhered to the SPIRIT statement. All data were stored and archived at the Data Center of DOT World Co., Ltd., Tokyo, Japan.

## Sample size calculation

Considering cystic lesions, the threshold for volume reduction was set at a minimum of 20%. This was based on the assumption that successful treatment should improve the conditions of a proportion of patients, achieving either complete response (target lesion resolved) or partial response (≥20% target lesion volume reduction). Limited data exist regarding magnetic

resonance imaging (MRI)-based evaluations for ethanolamine oleate in difficult-to-resect VMs. Kaji et al. [7] reported a median MRI-based lesion reduction rate of approximately 25% in 60 participants with VMs. Because the median corresponds to the 50th percentile of the data distribution, more than 50% of patients are expected to experience a reduction rate $\geq$ 20%, as shown in the figure presented by Kaji et al. A study conducted by Alexander et al. [25] reported on MRI-based reduction rates for VMs and demonstrated that at least 50% of participants achieved a reduction rate $\geq$ 20% [25].

It should be noted that Kaji et al. [7] calculated the lesion reduction rate from the measured area of the lesion. Although Kaji's method differs from that used in the current study, we considered the reduction rate reported in these previous studies to be suitable since the expected threshold and complete response and partial response values in the present study would be at least 50% [25].

Using a binomial test, a sample size of 19 patients is required to achieve 80% power under the alternative hypothesis that the response rate is expected to be 50%, with a threshold value of 20% and a target one-sided alpha of 2.5%. Considering potential dropout cases, the target number of patients with cystic lesions was set to 22. The same target was used for cases with diffuse lesions. The effect is studied in each group (cystic or diffuse type) separately. A total of 44 cases were targeted for recruitment.

## Participant inclusion criteria

Participants were selected according to the inclusion criteria set by investigators collecting patient information during outpatient consultations at each facility among the patients with VMs introduced at each facility. The inclusion criteria were any age; informed consent, either directly or through their legal guardians if under 20 years old; and VMs considered challenging to surgically remove, with sclerotherapy identified as the primary treatment option. Difficulty in resection was defined by the investigator or sub-investigator as a high risk of lifestyle dysfunction due to potential complications from excision or the risk of aesthetic impairment.

Moreover, patients needed to exhibit at least one target VM with a major axis diameter $\geq$ 30 mm in the extremities or $\geq$ 20 mm in the head and neck, as determined on MRI or computed tomography. Additionally, inclusion criteria stipulated the absence of a thrombus or organized tissue that could interfere with the evaluation of images or the judgment of treatment effectiveness in target VMs.

## Participant exclusion criteria

The exclusion criteria included multiple organ failure or disseminated intravascular coagulation; currently taking or recently started taking medications known to influence lesion resolution, such as propranolol or specific herbal medicines like Kamishoyosan or Ninjinyoeito; diabetes mellitus (HbA1c $\geq$8.0) or autoimmune disorders; Child–Pugh Class C liver dysfunction; renal dysfunction (estimated glomerular filtration rate <60 mL/min/1.73 m$^2$); or cardiac dysfunction (New York Heart Association Class $\geq$ 2).

Further exclusions encompassed patients who had undergone sclerotherapy within 6 months preceding signing of the informed consent, with a history of allergy to ethanolamine oleate or angiography contrast agents, and those who had undergone surgery exceeding 45 min within 2 weeks preceding signing of the informed consent. Additionally, participation in other clinical studies within 4 weeks preceding the signing of the informed consent, pregnancy, potential pregnancy, or lactation rendered individuals ineligible. Lastly, patients deemed ineligible by the investigator or sub-investigator were not enrolled.

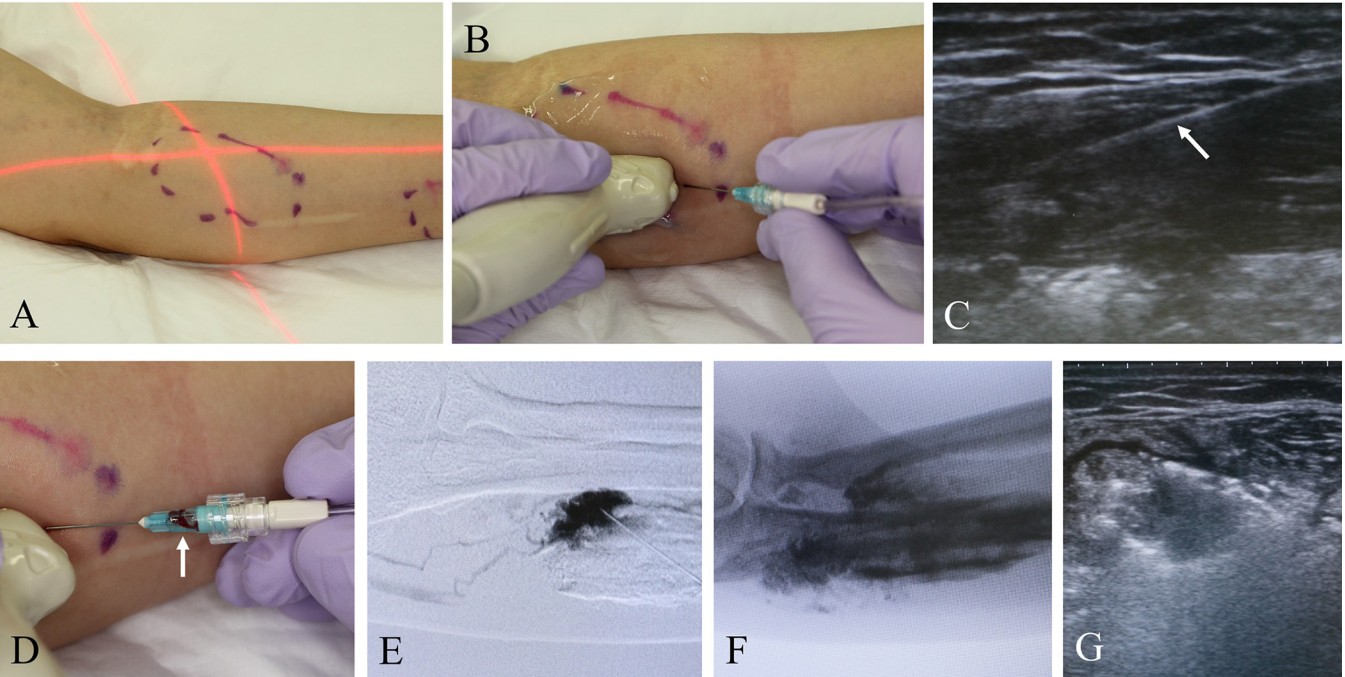

**Fig 2. Sclerotherapy procedure for venous malformations.** (A) Adjust the position of the fluoroscopy device. (B) Insert a needle under ultrasound guidance. (C) Ultrasound image at the time (B-mode). The inserted needle is visible (arrow). (D) Apply negative pressure to confirm the presence of back flow (arrow). (E) During digital subtraction angiography, contrast is injected to confirm the location of the lesion and outflow pathways. Once it is confirmed that the needle is in the appropriate position and the injection volume is verified, inject the sclerosing agent diluted with contrast. (F) Fluoroscopy after multiple iterations of this process showing the persistence of the sclerosing agent within the lesion due to the contrast mixture. (G) Ultrasound imaging confirming the presence of a sclerosing agent under ultrasound guidance.

## Intervention

The following procedures were performed by the investigator from each facility under general anesthesia in the operating room. Patients received injections of 5% ethanolamine oleate solution, double diluted with contrast or normal saline (Fig 2), with a maximum dose of 0.4 mL/kg. The same method of administration was used for children (<15 years old). The maximum volume of the prepared drug in one treatment was 30 mL. The safety of this dose is within the range approved for gastric variceal treatment in Japan [22], and its effect has been reported in previous studies [7, 13, 15–17]. The drug is administered only once; however, for patients with diffuse lesions, if there is little improvement in symptoms such as pain after 4 weeks and the principal investigator deems it necessary for treatment, additional dosing is permitted. In such cases, the dosage should be the same as the initial dose.

## Endpoints

The primary endpoint was achieving a $\geq 20\%$ VM volume reduction 3 months post-intervention. This endpoint was chosen based on the assumption that a positive correlation between VM symptoms and lesion volume translates to clinical effect [26]. MRI scans were outsourced to an external institution for lesion volume analysis. Afterwards, the images were separately evaluated by two independent board-certified radiologists without subject information using the image analysis software OsiriX (PixMeo SARL, Geneva, Switzerland) (Fig 3). The final value was determined by averaging the measurements obtained by each evaluator. The inter-observer correlation coefficient (2,1) was calculated to assess the reproducibility of the measurements between the evaluators.

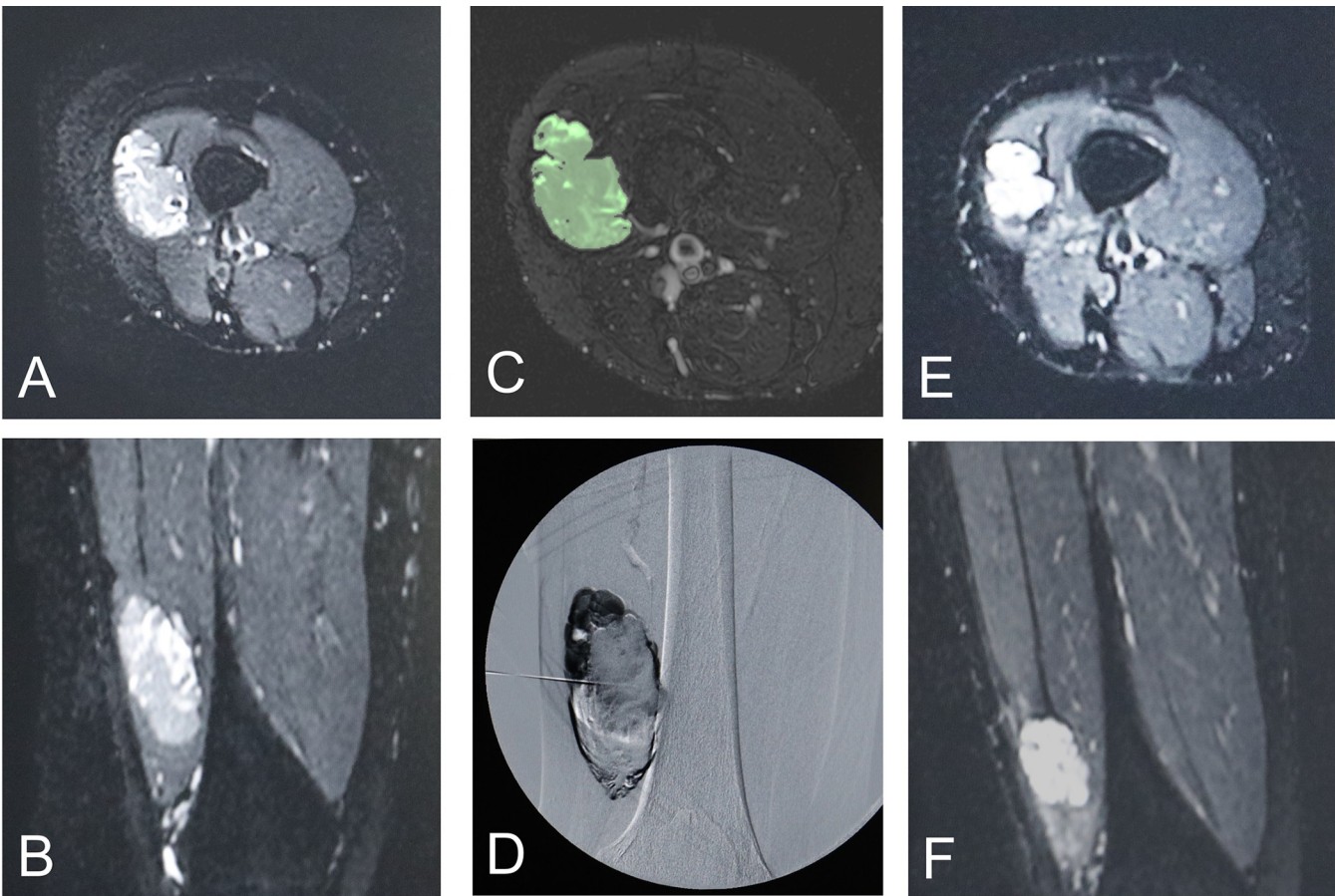

**Fig 3. A woman in her thirties with a painful intramuscular VM in the right thigh.** (A,B) Magnetic resonance imaging (MRI) before sclerotherapy (T2-weighted images: axial and coronal views). (C) One slide of the image subjected to segmentation using OsiriX (image analysis software) on MRI before sclerotherapy. (D) Digital subtraction angiography during sclerotherapy showing the administration of the sclerosing agent diluted with contrast. (E,F) MRI 3 months after sclerotherapy (T2-weighted images: axial and coronal views), confirming lesion volume reduction. The patient reported the resolution of pain at this time.

The key secondary endpoint was improvement in symptoms, particularly self-reported, lesion-associated pain scores, 3 months post-intervention. Patients aged 2 to 5 years were evaluated using the face scale (range: 1–6), developed by Wong-Baker [27], whereas those aged $\geq$ 6 years were evaluated using a visual analog scale (VAS) (range: 0–100). Two-time scales were assessed using the VAS: maximum pain within 24 h and maximum pain within 1 week.

Adverse events and adverse drug reactions were documented to evaluate safety endpoints. The severity of adverse events was determined based on the following criteria: grade 1 denoted mild signs or symptoms that were easily tolerable, grade 2 denoted moderate signs or symptoms that interfered with daily activities, and grade 3 denoted severe signs or symptoms that significantly impacted daily life.

## Statistical analysis of endpoints

The full analysis set comprised all patients who underwent the study intervention, ensuring a comprehensive evaluation of treatment outcomes. In contrast, the per-protocol set excluded patients deemed ineligible post-enrollment and those exhibiting severe protocol violations.

This approach aimed to uphold the integrity of the analysis by focusing on patients who adhered closely to the study protocol. The safety analysis set was identical to the full analysis set.

Primary analyses compared the number of patients who achieved ≥ 20% VM volume reduction 3 months after sclerotherapy relative to the baseline value using a one-sided binomial test with a significance level of $p < 0.025$ (one-sided). The 95% confidence intervals (CIs) for the proportion of patients achieving this reduction were estimated using the scoring method. This analysis was performed separately according to VM type.

Secondary analyses evaluated lesion-associated pain scores 3 months after sclerotherapy by estimating the median change from that at baseline with corresponding 95% CIs. The Wilcoxon signed-rank test was conducted to test the null hypothesis: "The median change from baseline to 3 months after sclerotherapy in the population was greater than 0." The significance level was set at $p < 0.025$ (one-sided). This analysis was performed separately according to VM type and patient age (<6 years and ≥6 years old). The occurrence of adverse events was assessed for safety evaluation.

All descriptive statistical analyses were performed using the SAS statistical software (version 9.4; SAS Institute, NC).

## Results

### Participant selection, baseline demographics, and clinical characteristics

Fig 1 illustrates the participant selection flowchart. Of the 45 patients who provided informed consent, one patient with cystic-type lesion was excluded owing to potential intracranial VM traffic during screening. The remaining 44 patients fulfilled the inclusion criteria. All patients were evaluated for the primary endpoint, and none of them discontinued treatment. Table 1 summarizes the baseline demographic and clinical characteristics of the patients. It is noteworthy that for both cohorts, the full analysis set, per-protocol set, and safety analysis set encompassed the same population under analysis for this study. This ensured consistency and comparability across the various analyses conducted throughout the research.

### Primary endpoint

All 44 patients underwent sclerotherapy, and none of the patients with diffuse VMs required additional treatment. Overall, a total of 26 patients (59.1%) achieved a VM volume reduction ≥ 20%, including 16 patients with cystic lesions (72.7%, 95% CI: 51.85–86.85%) and 10 patients with diffuse lesions (45.5%, 26.92–65.34%) S1 and S2 Tables. For both cohorts, the percentage of patients achieving a reduction rate ≥ 20% was statistically significant compared with the pre-specified criterion of 20% (cystic lesions: $p < 0.001$, diffuse lesions: $p = 0.001$). Data of patients who achieved a reduction rate ≥ 20% are shown in Table 2, and the percentage difference in lesion volume for each patient is shown in Fig 4. The interobserver correlation coefficient (2,1) for the MRI analysis was 0.999 (n = 88, 99.8–100%).

### Secondary endpoint

The key secondary endpoint was the difference in self-reported, lesion-associated pain scores from baseline to 3 months after sclerotherapy using the VAS or face scale. Significant improvements in VAS scores were observed for both cohorts (cystic lesions *vs.* diffuse lesions; 24-h maximum pain: n = 18, median: −2.0, $p = 0.014$ *vs.* n = 21, median: −7.0, $p = 0.013$; 1-week maximum pain: n = 13, median: −2.0, $p = 0.010$ *vs.* n = 20, median: −28.5, $p < 0.001$) S3 Table. Due to the limited sample size of patients aged ≤ 5 years, some statistical estimations and tests

**Table 1. Baseline demographics and clinical characteristics of the included patients.**

| | | Cystic Type n = 22 | Diffuse Type n = 22 | All n = 44 |
|---|---|---|---|---|
| **Age (year)** | Mean | 22.5 | 23.3 | 22.9 |
| | SD | 18.8 | 16.9 | 17.6 |
| | Min | 3 | 5 | 3 |
| | Median | 16.5 | 16.5 | 16.5 |
| | Max | 78 | 59 | 78 |
| **Sex (%)** | Male | 13 (59.1) | 9 (40.9) | 22 (50.0) |
| | Female | 9 (40.9) | 13 (59.1) | 22 (50.0) |
| **Body weight (kg)** | Mean | 45.73 | 49.02 | 47.37 |
| | SD | 24.32 | 15.60 | 20.26 |
| | Min | 14.7 | 20.3 | 14.7 |
| | Median | 49.1 | 50.5 | 49.1 |
| | Max | 92.0 | 75.2 | 92.0 |
| **Location of venous malformations (%)** | Trunk | 5 (22.7) | 4 (18.2) | 9 (20.5) |
| | Upper limb | 3 (13.6) | 8 (36.4) | 11 (25.0) |
| | Lower limb | 5 (22.7) | 10 (45.5) | 15 (34.1) |
| | Face | 5 (22.7) | 0 (0.0) | 5 (11.4) |
| | Oral | 1 (4.5) | 0 (0.0) | 1 (2.3) |
| | Neck | 3 (13.6) | 0 (0.0) | 3 (6.8) |
| **Previous treatment for venous malformations (%)** | Yes | 10 (45.5) | 12 (54.5) | 22 (50.0) |
| | No | 12 (54.5) | 10 (45.5) | 22 (50.0) |
| **Treatment history of sclerotherapy for target lesion (%)** | Initial treatment | 17 (77.3) | 13 (59.1) | 30 (68.2) |
| | Retreatment | 5 (22.7) | 9 (40.9) | 14 (31.8) |

Max, maximum; min, minimum; n, number; SD, standard deviation.

using the face scale-derived data could not be conducted. Differences in pain scores are presented in Table 3. In this study, for patients with diffuse lesions, provisions were made to allow additional treatment if improvement in symptoms such as pain was deemed insufficient. However, no patients required additional treatment.

## Safety evaluations

No cases of death or serious adverse events were reported. Adverse events of particular interest, defined as skin necrosis and visual impairment (for facial lesions) following

**Table 2. Patients who achieved ≥ 20% venous malformation volume reduction 3 months after the intervention.**

| Venous Malformations | No. of Patients | Achieving a Reduction of at Least 20% | | | P Value |
|---|---|---|---|---|---|
| | | No. of Achievers | Percentage | 95% Confidence Intervals | |
| **Cystic type** | 22 | 16 | 72.7 | 51.85–86.85 | <0.001 |
| **Diffuse type** | 22 | 10 | 45.5 | 26.92–65.34 | 0.001 |
| **All** | 44 | 26 | 59.1 | 44.41–72.31 | — |

The null hypothesis of this test: "The proportion of patients achieving ≥ 20% venous malformation volume reduction 3 months after sclerotherapy is < 20%."

The significance level of the test was 0.025 (on one side).

Test method: Test based on a binomial distribution.

Calculation method for confidence intervals: based on the score test.

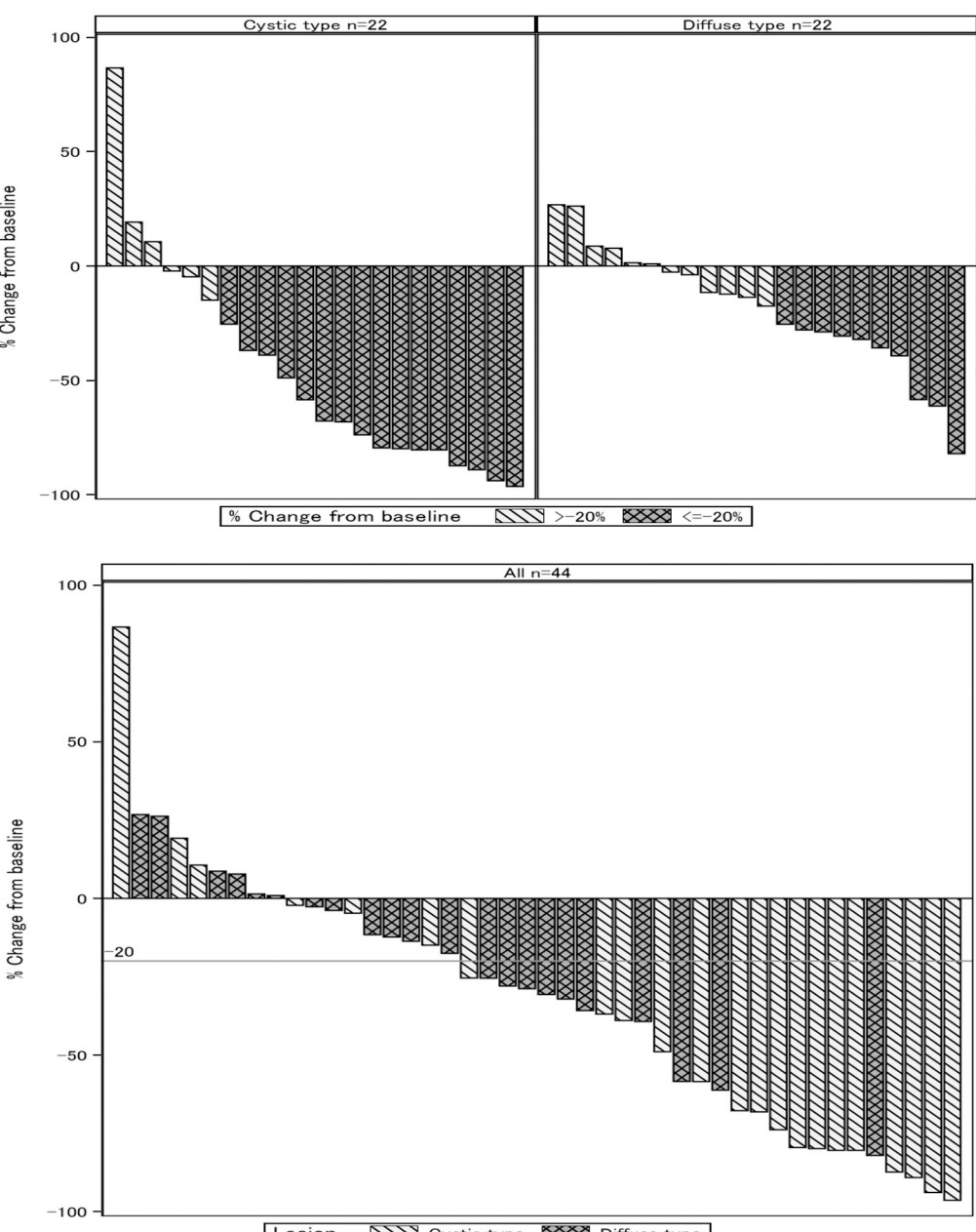

**Fig 4. Waterfall plot of the evaluated lesion volume.** (A) Diagrams of cystic lesions and diffuse lesions. (B) Diagrams for all cases.

administration, were not observed during the study period. Adverse events were documented in 42 patients (95.5%) S4 and S5 Tables. Among them, one patient experienced a grade 3 adverse event related to pain, while nine patients experienced grade 2 adverse events. These grade 2 events included three cases of pain; two of hemoglobinuria; and one each of swelling, increased fibrin D-dimer, increased serum creatinine phosphokinase, and ulnar nerve paralysis. Additionally, 32 patients reported grade 1 adverse events. Notably, hemoglobinuria occurred in 23 patients (52%), including nine with cystic lesions (40.9%) and 14 with diffuse lesions (63.6%). All cases of hemoglobinuria occurred on the day of drug administration, and

**Table 3. Changes in self-reported, lesion-associated pain scores 3 months after sclerotherapy.**

| Venous Malformations | Scale | Age | Type of Pain | No. of Patients | Median Change From Baseline | 95% Confidence Intervals | p Value |
|---|---|---|---|---|---|---|---|
| **Cystic type** | VAS | ≥ 6 years | Maximum pain in 24 h | 18 | −2.0 | −25.00 to 0.00 | 0.014 |
| | | | Maximum pain for 1 week | 13 | −2.0 | −51.00 to 0.00 | 0.010 |
| | Face scale | 2–5 years | Maximum pain in 24 h | 2 | 0.0 | 0.00 to 0.00 | — |
| | | | Maximum pain for 1 week | 2 | −0.5 | −1.00 to 0.00 | 0.500 |
| **Diffuse type** | VAS | ≥ 6 years | Maximum pain in 24 h | 21 | −7.0 | −31.00 to −1.00 | 0.013 |
| | | | Maximum pain for 1 week | 20 | −28.5 | −34.00 to −16.00 | <0.001 |
| | Face scale | 2–5 years | Maximum pain in 24 h | 1 | 0.0 | — | — |
| | | | Maximum pain for 1 week | 1 | 0.0 | — | — |

The null hypothesis of the test: "The median change in self-reported, lesion-associated pain scores from baseline to 3 months after sclerotherapy is ≥ 0."

The significance level of the test was 0.025 (on one side).

Method of testing: Wilcoxon signed-rank test.

Confidence intervals were calculated according to the method described by Hahn and Meeker [28].

haptoglobin therapy was initiated on the same or subsequent day. The administration of haptoglobin successfully prevented renal impairment associated with hemolysis in all patients, and of 23 cases, hemoglobinuria disappeared within 3 days in 21 cases and within 1 month in 2 cases.

## Discussion

### Strengths of this study

This clinical trial is the first well-designed study to provide robust evidence regarding the effect and safety of ethanolamine oleate for patients with difficult-to-resect VMs. The rarity of VMs makes it challenging to conduct large-scale trials, and the lack of clinically robust evidence hinders the pharmaceutical approval of ethanolamine oleate in some countries, including Japan [11]. This high-quality study, demonstrating positive outcomes, has the potential to pave the way for regulatory approval in such regions. Patients affected by VMs face great challenges due to the rarity of their disease and inadequate national-level support; therefore, this study holds promise for the global recognition of ethanolamine oleate treatment.

### Interpretation of the study endpoints

Regarding the primary endpoint, the majority of patients achieved ≥ 20% VM volume reduction 3 months after sclerotherapy. Specifically, 16 patients (72.7%) with cystic lesions and 10 patients (45.5%) with diffuse lesions demonstrated a volume reduction ≥ 20%. These findings suggest the utility of ethanolamine oleate administration in reducing VM volume for both cystic and diffuse lesions, consistent with previous reports [7, 25]. Given that thrombus formation and lesion enlargement can induce diverse symptoms in patients with VMs, a volume reduction ≥ 20% is clinically significant. As such, patients who achieved this reduction have reported improvements in symptoms such as pain.

Regarding the secondary endpoint, significant improvements in self-reported, lesion-associated pain scores were observed 3 months after sclerotherapy, which aligned with the findings of previous studies [14, 21, 29–32]. Although the mechanism through which pain is alleviated by sclerotherapy for VM is not currently well understood, the following mechanisms are hypothesized. Upon injection into the blood vessel, the sclerosing agent damages the endothelium, causing the vessel to contract, which obstructed the blood flow. This may reduce

abnormal blood flow and pressure, potentially alleviating pain. Additionally, the sclerosing agent can strengthen the vessel walls, reducing their leakage and dilation, which can also alleviate pain. Furthermore, the reduced frequency of thrombus formation decreases the likelihood of thrombophlebitis, contributing to pain reduction.

## Safety of this method

Regarding safety, our study observed a high rate of adverse effects; however, most of these were attributed to temporary inflammation directly related to the pharmacological effects of sclerotherapy. Overall, the safety of the treatment can be considered to be reasonably assured. The adverse effect that warrants attention is hemoglobinuria, which was reported in approximately 52% of the patients in our study. This was comparable to the prevalence reported in previous Japanese clinical observational studies (27.7–50.0%) [7, 14, 33]. In cases where hemoglobinuria occurs during sclerotherapy with ethanolamine oleate, there is a potential risk of progression to acute renal failure [34]. Therefore, clinicians should be aware of the importance of early administration of haptoglobin, a plasma fraction preparation [35]. In a study by Fujiki et al. [33], early haptoglobin administration for hemoglobinuria prevented renal damage in all cases, and these authors identified that a 5% ethanolamine oleate dose $\geq 0.18$ mL/kg was a predictive factor for gross hematuria. To avoid acute renal failure, it is essential to disseminate information regarding the risk of hemoglobinuria with ethanolamine oleate use through guidelines or similar resources. Interestingly, no cases of skin ulceration or skin necrosis were observed in the present study, whereas previous studies have reported incidence rates of 3.0–14.7% [7, 21, 36]. The absence of these complications implies a safer treatment profile with our regimen.

## Limitations of this study

Some limitations were noted in the present study. First, the study employed a single-arm design and included a limited sample size. Although this reflected the rarity of the studied disease, this also limits the generalizability of our findings. Nevertheless, our study population was comparable to that included in the trial that served as the basis for approval (44 patients), and our study employed a post-intervention evaluation period of 3 months. Larger scale studies with longer follow-up periods are needed to clarify our findings. Second, while no age restrictions were imposed in this study, we were unable to recruit children $\leq 2$ years of age, which is a significant population affected by VMs. Thirdly, the inclusion of some patients who lacked baseline pain assessment data and the presence of age-related variations in pain scale comprehension limited the applicability of a single pain assessment tool. Finally, this study did not include a placebo group, such as normal saline injection. Although the effectiveness of sclerotherapy for VMs is well documented in numerous guidelines and textbooks and is widely recognized in the medical field, there had been no high-evidence-level clinical research reports available previously. This is why this study was conducted. Prior to initiating this study, we consulted with the Pharmaceuticals and Medical Devices Agency (PMDA), which oversees pharmaceutical regulations in Japan, regarding the protocol and were advised that including a placebo group was unnecessary. Furthermore, since this treatment is performed under general anesthesia during hospitalization and VM is progressive, it was ethically challenging to establish a placebo group.

Subsequently, we examined various types of bias. First, regarding selection bias, even though some cases had predominantly diffuse lesions, if the target lesions were cystic, they were classified as part of the cystic group. This might mean that pain assessments in the cystic group could be influenced by the diffuse lesions. Regarding detection bias, since the evaluation

of lesion volume was conducted independently by two external evaluators who were not informed of the subject information, it is expected that the impact of this bias was minimal. Regarding performance bias, although the same technique of sclerosing agent injection was used for both cystic and diffuse lesions, there might be variability in the amount of sclerosing agent injected owing to uncertain areas in diffuse lesions. However, since this study was not a comparative trial, the influence of this bias on the research results is not considered to be problematic.

## Conclusion

To the best of our knowledge, this is the first prospective clinical trial to expound the indications for ethanolamine oleate administration in patients with difficult-to-resect VMs in Japan. In our study population of 44 patients, ethanolamine oleate significantly reduced VM lesion volume and improved pain. Regarding safety, clinicians should be aware of the significance of early haptoglobin administration for drug-induced hemoglobinuria. Our results highlight the potential of ethanolamine oleate as a treatment option for patients with difficult-to-resect VMs. Further studies with larger sample sizes and longer follow-up durations are warranted to evaluate the long-term clinical outcomes of this approach.

## Supporting information

**S1 Protocol. Protocol.**
(DOCX)

**S2 Protocol. Protocol Appendix 1.**
(DOCX)

**S3 Protocol. Protocol Appendix 2.**
(DOCX)

**S1 Table. Reduction rate (%) of venous malformations.**
(PDF)

**S2 Table. Target lesion volume.**
(PDF)

**S3 Table. Pain scores associated with target lesion.**
(PDF)

**S4 Table. All adverse events.**
(PDF)

**S5 Table. Adverse drug reactions.**
(PDF)

**S1 File. TREND checklist.**
(PDF)

## Acknowledgments

This study was supported by the Kobe Clinical and Translational Research Centre. We thank all the staff for their involvement in this clinical trial.

## Author Contributions

**Conceptualization:** Mine Ozaki, Tadashi Nomura, Yasumasa Kakei, Keiko Miyakoda, Naoko Kashiwagi, Takahiro Yasuda, Tsuyoshi Kaneko.

**Data curation:** Keiko Miyakoda.

**Formal analysis:** Junko Ochi, Shimpei Akiyama, Yasumasa Kakei, Keiko Miyakoda, Kiyoko Kamibeppu, Takafumi Soejima.

**Funding acquisition:** Mine Ozaki.

**Investigation:** Mine Ozaki, Tadashi Nomura, Keigo Osuga, Masakazu Kurita, Ayato Hayashi, Shunsuke Yuzuriha, Noriko Aramaki-Hattori, Makoto Hikosaka, Yuki Iwashina.

**Methodology:** Mine Ozaki, Tadashi Nomura, Taiki Nozaki, Yasumasa Kakei, Takahiro Yasuda.

**Project administration:** Mine Ozaki, Tadashi Nomura.

**Resources:** Mine Ozaki, Tadashi Nomura, Keigo Osuga, Masakazu Kurita, Ayato Hayashi, Shunsuke Yuzuriha, Noriko Aramaki-Hattori, Makoto Hikosaka, Taiki Nozaki, Michio Ozeki.

**Supervision:** Mine Ozaki, Tsuyoshi Kaneko, Kiyonori Harii.

**Validation:** Mine Ozaki, Michio Ozeki, Yasumasa Kakei.

**Writing – original draft:** Mine Ozaki, Yasumasa Kakei.

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
