## [Decision Letter · Decision Letter 0]

24 Jul 2024

PONE-D-24-12872Efficacy and Safety of Ethanolamine Oleate in Sclerotherapy in Patients with Difficult-To-Resect Venous Malformations: A Multicenter, Open-Label, Single-Arm StudyPLOS ONE

Dear Dr. Ozaki,

Thank you for submitting your manuscript to PLOS ONE. After careful consideration, we feel that it has merit but does not fully meet PLOS ONE’s publication criteria as it currently stands. Therefore, we invite you to submit a revised version of the manuscript that addresses the points raised during the review process. Please submit your revised manuscript by Sep 07 2024 11:59PM. If you will need more time than this to complete your revisions, please reply to this message or contact the journal office at plosone@plos.org. Please include the following items when submitting your revised manuscript:A rebuttal letter that responds to each point raised by the academic editor and reviewer(s). You should upload this letter as a separate file labeled 'Response to Reviewers'.A marked-up copy of your manuscript that highlights changes made to the original version. You should upload this as a separate file labeled 'Revised Manuscript with Track Changes'.An unmarked version of your revised paper without tracked changes. You should upload this as a separate file labeled 'Manuscript'.

We look forward to receiving your revised manuscript.

Kind regards,

Sonam Khurana

Academic Editor

PLOS ONE

Journal Requirements:

"This research was supported by the Japan Agency for Medical Research and Development (AMED), Grant Number JP20lk0201115."

3. In the online submission form, you indicated that "Requests should be directed to the corresponding author by email."

Reviewers' comments:

Reviewer's Responses to Questions

**Comments to the Author**

1. Is the manuscript technically sound, and do the data support the conclusions?

Reviewer #1: Partly

Reviewer #2: Yes

2. Has the statistical analysis been performed appropriately and rigorously? 

Reviewer #1: Yes

Reviewer #2: Yes

3. Have the authors made all data underlying the findings in their manuscript fully available?

Reviewer #1: No

Reviewer #2: Yes

4. Is the manuscript presented in an intelligible fashion and written in standard English?

Reviewer #1: Yes

Reviewer #2: Yes

5. Review Comments to the Author

Reviewer #1: The authors reported about the results of prospective, single arm multicenter clinical trial to explore the effect and safety of sclerotherapy in patients with difficult-to-resect venous malformations treated with ethanolamine oleate. They concluded that ethanolamine shows potential as a therapeutic sclerosing agent for patients with difficult-to-resect venous malformations.

In general, this is a nice study but the presentation need some review from the methodological side.

To be more specific:

• Data sharing would be informative in particular for the review process to being able to follow the authors arguments.

• Please go through the manuscript and correct for the use of the word efficacy. Usually "efficacy" is associated estimation of the treatment effect in a comparative trial. Here the word "effect" - meaning estimation of the effect of a treatment - should be used.

• The term "open label" is rather obvious in a single arm trial and can be deleted.

• There are two groups considered resulting in the same sample size. This seem to be surprising! Please make clear whether a stratified sampling in the two groups (cystic type or diffuse) is applied. This should be stated clearly, in particular which number of patients in the subgroups must be screened for inclusion (see figure 1). Figure 1 has to be modified in this case.

• As the group effect seem not to be interest, I assume the logic of the design is to estimated the overall (pooled) effect across both groups. Presentation of subgroup results might be of second line interest. Please explain the logic of the study design.

L64: Please clarify whether this is a pediatric trial or a trial in an adult and pediatric population.

L68: The term "changes" is unclear at this place of the paper.

L141: please delete open-label and include single arm

L165: The information of the sample size justification is somewhat confusing. What I understood the effect under the null hypothesis is assumed to be a proportion of 20% which should be distinguished from a relevant effect of 50% (Alternative). To show this in a single arm trial (at least 80% Power, 2.5% one-sided significance level) using a binomial test a sample size of 19 is necessary. The effect is studied in each group (cystic type or diffuse) separately. Please provide some text like this.

L248: Definition of the primary endpoint is not aligned with the text in L68/69 where two endpoints are mentioned.

L279: Analysis set description doesn ot need extensive text. This paragraph should be summarized in one sentence.

L426: I do not like layman expressions like "relatively limited sample size". The term "relatively" is confusing.

L424: I see some other limitations with the study:

• Please argue why a comparative study of ethanolamine against placebo is not possible or considered here.

• Please elaborate the influence of bias (allocation bias, detection bias, performance bias) on the study results.

Reviewer #2: The authors present an interesting study looking at the efficacy and safety of ethanolamine oleate in sclerotherapy for patients with difficult to resect AVM and found that 59.1% of patients had a >20% volume reduction of AVM with a single dose of ethanolamine oleate solution.

How were these lesions determined to be "difficult to resect"? Was this based on an interdisciplinary team decision or a single provider's gestalt?

Is Ethanolamine oleate FDA approved for this indication? If not, this will be difficult to gain any traction clinically in the United States.

The patient demographics should have statistical analyses performed to demonstrate that there were no significant differences between the cystic and diffuse cohort.

Did any patients require resection following the ethanolamine injection? How many required repeated injections to maintain these results?

A 50% adverse effect rate is quite high for any sort of medication/therapy...how do the authors propose this could be avoided or is there another way to preemptively avoid this adverse effects? With such a high adverse effect rate, this relates back to my question above regarding FDA approval for this indication and if this would ever pass regulatory approvals into daily clinical practice.

The authors need to include controls - patients with difficult to resect AVMs that only receive saline injection and demonstrate that a similar effect is not seen with saline injections alone. Perhaps the tumescence from the volume of the injection is enough to cause the AVM to vasospasm and shrink.

6. PLOS authors have the option to publish the peer review history of their article (what does this mean?). If published, this will include your full peer review and any attached files.

Reviewer #1: No

Reviewer #2: No

---

## [Author Response · Author response to Decision Letter 0]

6 Sep 2024

Thank you for reviewing our manuscript. We have carefully considered your valuable feedback and have provided point-by-point responses for your kind review.

---

## [Decision Letter · Decision Letter 1]

24 Nov 2024

Effect and Safety of Ethanolamine Oleate in Sclerotherapy in Patients with Difficult-To-Resect Venous Malformations: A Multicenter, Single-Arm Study

PONE-D-24-12872R1

Dear Dr. Ozaki,

We’re pleased to inform you that your manuscript has been judged scientifically suitable for publication and will be formally accepted for publication once it meets all outstanding technical requirements.

Kind regards,

Wenbin Tan

Academic Editor

PLOS ONE

Additional Editor Comments (optional):

Reviewers' comments:

Reviewer's Responses to Questions

**Comments to the Author**

Review Comments to the Author

Reviewer #1: All my previous comments are adressed, I do not have any further comments. I consider the paper as ready for publication.

Reviewer #3: The revised manuscript was good written, designed and discussed. All comments of reviewers have been addressed.

---

## [Editor Report · Acceptance letter]

13 Dec 2024

PONE-D-24-12872R1 

PLOS ONE

Dear Dr. Ozaki, 

I'm pleased to inform you that your manuscript has been deemed suitable for publication in PLOS ONE. Congratulations! Your manuscript is now being handed over to our production team.

Kind regards, 

on behalf of

Dr. Wenbin Tan 

Academic Editor

PLOS ONE